# Ecological Risk Assessment of Trace Metal in Pacific Sector of Arctic Ocean and Bering Strait Surface Sediments

**DOI:** 10.3390/ijerph19084454

**Published:** 2022-04-07

**Authors:** Juan Wang, William A. Gough, Jing Yan, Zhibo Lu

**Affiliations:** 1Department of Environmental Science and Engineering, Tongji University, Shanghai 200092, China; wangjuan@tongji.edu.cn (J.W.); yanjingagnes@163.com (J.Y.); 2State Key Laboratory of Pollution Control and Resource Reuse, Tongji University, Shanghai 200092, China; 3Department of Physical and Environmental Sciences, University of Toronto Scarborough, 1265 Military Trail, Toronto, ON M1C 1A4, Canada; william.gough@utoronto.ca

**Keywords:** trace metals, surface sediment, ecological risk assessment, enrichment factor, geo-accumulation index, potential ecological risk index

## Abstract

The arctic region is a remote area with relatively few anthropogenic inputs, but there is increasing concern over toxic trace metal contamination in the Arctic Ocean. In this study, distribution characteristics of eight trace metals in the surface sediment of the Pacific Sector of the Arctic Ocean and Bering Strait are analyzed. The geochemical baseline value of each metal element is explored using the relative cumulative frequency curve method; the enrichment factor (*EF*), geo-accumulation index (*I_geo_*) and potential ecological risk index (*RI*) are applied to assess the ecological risk level of the trace metals. According to the results, Cu, As and Hg had a little more content variation, and their contents in some areas were significantly higher than the previous reports. *EF* values show an obvious enrichment of element As, followed by Cr element with the moderate enrichment; the enrichment of the other six elements are not related to human activity. The *I_geo_* value shows a moderately contaminated to heavily contaminated level of As and a moderately contaminated level of Cr. According to the potential ecological risk indexes in each site, most sites are at a low ecological risk level except five sites with RI/baseline values exceeding 150 which are at a moderate ecological risk level.

## 1. Introduction

Environmental pollution by trace metals is a serious problem worldwide due to their toxicity and persistence. In 2014, the United States Environmental Protection Agency (USEPA) classified 83 priority metal pollutants in terms of hazard, exposure levels, persistence and bioaccumulation, which included arsenic (As), cadmium (Cd), chromium (Cr), mercury (Hg), nickel (Ni) and lead (Pb) with high hazard scores, in combination with 345 different chemical substances. For other metals, such as copper (Cu) and zinc (Zn), which have not yet been seen as threats to human health, toxicological studies are resulting in the gradual recognition of new hazards [1].

The trace metals in seawater can interact with suspended particles via adsorption, complexation and precipitation and then be transferred to and become enriched in surface sediments [2]. In some conditions, trace metals in sediment can be released to the overlying seawater, causing the “secondary pollution” of seawater [3]. Sediments are recognized as the main reservoir and source of metals in the marine environment, and they play an important role in transporting and accumulating potentially harmful trace metals [4]. Most trace metals cannot be biodegraded and directly threaten human health through food chain transmission [5,6].

The ecological risk assessment of the levels of trace metals in coastal ecosystems has attracted a lot of attention from both researchers and policymakers owing to their potential toxicity, persistence, tendency to bio-accumulate and public concern for seafood safety [7,8].

The polar regions have become one of the regions where climate change and human impacts are most pronounced. Potential environmental pollution in the polar regions has become a major challenge to local marine ecosystems. The arctic region is a remote area with relatively few anthropogenic inputs, but there is increasing concern over toxic trace metal contamination in the Arctic Ocean. Many studies have shown that trace metal concentrations in Arctic marine sediments are subjected to anthropogenic contamination to different extents [9,10,11,12,13,14,15]. Furthermore, global warming may cause exacerbating effects such as changes in biological processes, increased corrosion in river basins and permafrost melting, leading to increased metal accumulation in Arctic marine sediments. The Bering Strait is the only conduit of water masses and organic matter between the North Pacific and Arctic Oceans. Metal accumulation in sediments in this area may directly threaten large numbers of arctic marine invertebrates, fish, birds and mammals [16,17], possibly affecting the entire Arctic ecosystem. Consequently, there is an urgent need to assess the sediment-related ecological risk in this region.

In this study, based on the total concentration of eight trace metals (As, Cd, Cu, Cr, Hg, Pb, Zn and Ni) from 64 surface sediment samples in the Arctic Ocean, the spatial distribution of trace metals was analyzed, the geochemistry baseline value of each metal element was explored and three methods were chosen to assess the ecological risk level of trace metals.

## 2. Materials and Methods

### 2.1. Study Area

The study area is located in the Bering Strait and Pacific sector of the Arctic Ocean, between the Eurasian and North American continents. The Bering Strait is one of the largest marginal seas worldwide, with an area of 2.29 × 10^6^ km^2^ and volume of 3.75 × 10^6^ km^3^, and it is the only channel between the Bering and Chukchi Seas. Pacific seawater flows through the Bering Strait into the Arctic Ocean. The strait is relatively shallow, with maximum and average depths of 200 and 42 m, respectively. To the north of the Strait is the Canadian Arctic shelf, with a water depth > 4000 m, and to the south is the Aleutian Basin with an average water depth of 3000 m. The profile of the Bering Sea itself has been likened to the Grand Canyon (USA). The Chukchi Shelf is relatively shallow (50 m depth) and flat, with small areas of glacial plain. Most near shore terrigenous sediments were carried by the Kobuk and Noatak rivers, while the sediments of the central Chukchi Shelf were carried mainly by the Yukon River.

### 2.2. Sample Collection and Preparation

Surface sediment samples were collected using box-core. The subsamples for analyses were taken from the top 5 cm of the box center. Each bulk sample was divided into two parts for the determination of elements and the analysis of grain size.

Surface sediment samples were collected from the study area during the 4th (July–August 2010, *n* = 29), the 5th (July–September 2012, *n* = 37) and the 6th (July–August 2014, *n* = 20) Chinese National Arctic Research Expedition (CHINARE). Sampling locations are shown in Figure 1. 

All sampling tools were washed in local seawater, high purity acetone, and dichloromethane before sampling. Sediment samples were stored in acid-washed glass bottles at −20 ℃ before analysis.

The sampling sites of the 4th Arctic Scientific Expedition of China are mainly located in the continental shelf of the Bering Strait and the North Ocean deep basin, including the Markelov basin, the Alpha ridge, the Northwind ridge, the Beaufort Sea and the Canadian Sea Basin in northern Alaska. The sampling sites in the 5th Arctic Scientific Expedition ran through the Bering Strait, covering the Bering Sea shelf area, the Bering Strait and Chukchi Shelf area.

Due to the data availability and comparability, in the following study, the analyses of arsenic and mercury are based on the 66 samples collected in the 4th and 5th CHINARE, and the other elements (cadmium, chromium, copper, nickel, lead and zinc) are based on the 86 samples collected in the 4th, 5th and 6th CHINARE.

### 2.3. Laboratory Analysis of Total Concentration of Trace Metals

Sediment samples for chemical analysis were stored in glass bottles and the bottles are stored in low density polyethylene (LDPE) containers and transported with dry ice at −20 °C until analysis.

The detection procedures for seven elements (arsenic, cadmium, chromium, copper, nickel, lead and zinc) can be summarized as follows: the sediment samples were pretreated by drying in air at room temperature, crushing, sieving and grinding to less than 200 mesh. Digestion of the samples was carried out to destroy organic matter and to dissolve suspended solids. An Automatic Microwave Digestion System (ETHOSE, Italy MileStone, SRL) was used for the digestion of the samples, and this equipment converts elements into a single high valence state or into inorganic compounds that are easy to separate. Representative 0.5 g (dry weight) samples were digested in 10 mL of concentrated nitric acid in a Teflon (R) PFA vessel or equivalent for 10 min using microwave heating. The temperature should rise to 180 °C in approximately 15 min and remain at 180 °C for 20 min. After cooling, the samples were either placed in a centrifuge or allowed to settle by gravity overnight and filtered to remove insoluble material. The samples were then diluted and detected with an inductively coupled plasma mass spectrometer (ICP–MS 7700, Agilent, Santa Clara, CA, USA). The detection limits for As, Cd, Cr, Cu, Ni, Pb and Zn were 0.02, 0.04, 0.01, 0.03, 0.01, 0.01 and 0.06 ppb, respectively. The content of total mercury was detected with Direct Mercury Analysis (DMA-8, MileStone, Milano, Italy).

All the acids used in this study were guaranteed reagents, and no laboratory contamination was detected during the analyses because the concentrations in the reagent blanks were below the instrumental limits of detection.

Replicate analyses were carried out on 3 groups after digestion, and all the analytical results are average values of the replicates. Experiments showed that the standard method of sample collection, the pretreatment and the analytical methods yield accurate data.

### 2.4. Geochemical Baseline Value Determination Based on Relative Cumulative Frequency

Bauer and Bor [18] developed the relative cumulative frequency method that uses normal decimal coordinates. Taking the element concentration as the *X*-axis, and the relative cumulative frequency (for example, a total of 100 numbers, each of which accounts for 1% of the total, sort from the smallest concentration to the largest, the first number accounts for 1% of the total, plus the second number accounts for 2%… plus the 99th number accounts for 99%, plus the 100th number accounts for 100%) as the *Y*-axis, then the relative cumulative frequency curve is established. Generally, there are two inflection points in the curve, the lower point may represent the upper limit of the natural source concentration of the element (baseline range). If the value of the lower point is less than the average or median value of the sample element concentration, it can be regarded as a pure natural source, the average value of the baseline range can be used as the baseline value. The upper point may represent the abnormal lower bound caused by human activity, and the area between the two inflection points may or may not be relevant to human activity. If the curve is approximately linear, the content of the sample itself may represent the background range (baseline). Sometimes an inflection point occurs when the concentration is very low and close to zero, which indicates that the data sources are unreliable, a re-test should be taken to ensure that there is no data below the detection line.

### 2.5. Ecological Risk Assessment Methods

#### 2.5.1. Enrichment Factor (*EF*) Analysis

*EF* is an important indicator that quantitatively assesses the levels and sources of trace metal contamination [19]. Metal concentrations were normalized to the textural characteristic of sediments with respect to Fe. Iron was selected because it is a major sorbent phase for trace metals and is a quasi-conservative tracer of the natural metal-bearing phases in fluvial and coastal sediments [20].

*EF* is calculated as displayed below:*EF* = (*C_i_*/*C_Fe_*) *_samples_*/(*B_i_*/*B_Fe_*) *_earth crust_*(1)
where *EF* is the enrichment level of a certain trace metal, *C_i_* is the measured concentration of *i* trace metals in the sediment, *C_Fe_* is the measured concentration of the *Fe* in the sediment, *B_n_* and *B_ref_* are the concentration of *i* and *Fe* in the earth crust, respectively (Table 1) [21].

Table 2 is the classification standard of *EF* value.

#### 2.5.2. Geo-Accumulation Index (*I_geo_*) Analysis

Geo-accumulation index (*I_geo_*) was introduced by Muller [22] to assess trace metal accumulation in sediments and to evaluate the degree of metal pollution in the surface sediment [23]. This index is used by many scientists to evaluate the level of contamination of individual metals in sediments.

*I_geo_* is calculated as displayed below:(2)$$I_{geo}=\log_{2}\left[\frac{C_{n}}{1.5B_{n}}\right]$$
where *Cn* is the measured concentration in the sediment for the metal *n*, *Bn* is the background value for the metal *n*, and the factor 1.5 is used because of possible variations of the background data due to lithological variations. In this study, the crustal abundance data from Turekian (Table 1) and the geochemical baseline value of the study area calculated in this study as seen in Section 3.2 are selected to be the background concentration.

The geo-accumulation index is classified into seven categories (0–6), ranging from uncontaminated to extremely contaminated (Table 3) [24].

#### 2.5.3. Potential Ecological Risk Index

The potential ecological risk index method was proposed by Hakanson based on principles of sedimentology to assess the characteristics and environmental behaviors of trace metal contaminants in coastal sediments [25]. This method comprehensively considers the synergy, toxicity, concentration and ecological effect of trace metals. The Er and *RI* are introduced to assess the ecological risk of a single trace metal element and the comprehensive potential ecological risk, respectively, and are calculated as displayed below:(3)$$\begin{array}{c} C_{f}^{i}=C_{D}^{i}/C_{R}^{i}\\ E_{R}^{i}=T_{R}^{i}\times C_{f}^{i}\\ RI=\overset{m}{\underset{i=1}{\sum }}E_{R}^{i} \end{array}$$
where $C_{D}^{i}$ is the measured concentration of trace metal in each sampling point; $C_{R}^{i}$ is reference value, here the geochemical baseline value of the study area calculated in this study as seen in Section 3.2, and the preindustrial reference values of trace metals from Hakanson [25] are selected (Table 4); $C_{f}^{i}=C_{D}^{i}/C_{R}^{i}$ is the pollution of a single element factor; $E_{R}^{i}$ is the potential ecological risk index of a single element; *RI* is a comprehensive potential ecological risk index; and $T_{R}^{i}$ is the biological toxic factor of a single element, which is determined for As = 10, Cd = 30, Cr = 2, Cu =5, Ni = 5, Pb = 5, Zn = 1 and Hg = 40 [25].

Table 5 lists the relationship among *RI*, $E_{R}^{i}$ and risk levels.

## 3. Results

### 3.1. Concentration Level of Trace Metals and Spatial Distribution in Study Area

Table 6 is the statistical distribution of eight trace metals in study area. The order of trace metal content variation in surface sediments from high to low is Cu > Hg > As > Cd > Ni > Cr > Zn > Pb, in which the concentration changes in Cu, Hg and As exceeding one order of magnitude. The skewness coefficient shows that the distribution of other trace metals is right deviation except Zn.

Compared with previous studies on the trace metal concentration in this area, the content of Cr and Cu in some sites of the Bering Sea in this study is significantly higher than that of previous reports, the content of Pb shows a general higher trend, while the content of Cu and As of several sites in the Chukchi continental shelf and the northern area is significantly higher than the previous reports [9,11,13,26].

Figure 2 shows the comparison of trace metal contents in this study with other Arctic areas in other literatures [10,14,27,28,29,30].

Compared with the trace metal content in other Arctic Ocean areas (Figure 2), it can be seen that the content of As is at a slightly lower level, similar with that in the Kara Sea, Laptev Sea, East Siberia Sea and Beaufort Sea. The content of Cd is at a higher level compared with the other sea areas, slightly below Svalbard and Beaufort Sea, and may have accumulation to some degree. The content of Cr is lower than other sea areas in the Arctic Circle as a whole. The content of Cu is obviously lower than that of the Greenland Sea, and a little higher than that of the Pechora Sea and the East Siberia Sea. The content of Ni is higher than that in the other sea areas, except for the Greenland Sea. The content of Pb is generally lower than that of the other sea areas. The content of Zn and Hg is higher than that of Pechora, like other sea areas.

In order to facilitate the analysis, the study area is subdivided into seven regions according to the terrain and the sediment environment, which are the south Bering Sea, the Bering Sea shelf, the southern entrance of the Bering Strait, the south Chukchi Sea shelf, the outer shelf of Chukchi, the Beaufort Sea to the Canadian Basin near the northwest of Alaska, as well as the deep-sea area of the Arctic Ocean (followings are the same).

The minimum content of As is located in the station of B02 in the southwest Bering Sea near the Aleutian Islands (the 4th CHINARE, 2.98 mg·kg^−1^). The highest content is BN09 (the 4th CHINARE, 38.34 mg kg^−1^) in the deep sea of the Arctic Ocean, which is obviously higher than the sites nearby. The content of As gradually rises from the Beaufort Sea to the Canadian Basin and is generally higher than those of the Chukchi Sea. The content of As in the deep-sea of the Arctic Ocean is generally higher than those in the Bering Sea and the Chukchi Sea area (Figure 3a).

The minimum content of Cd is located at the MOR02 station (the 4th CHINARE, 0.05 mg·kg^−1^) at the junction of the south of the Canadian Basin and the Northwind ridge, and the highest content is located at the SR12 station (the 5th CHINARE, 0.47 mg kg^−1^) on the edge of the Chukchi Shelf. For the Bering Sea area, Cd content rises eastward from the central and western side to the highest at BS02 station, and then gradually drops eastward to the lowest near the Alaskan coast. The Cd content at the southern entrance of the Bering Strait is lower than other areas. The Cd content in the southern and northern Chukchi Sea varies greatly, and the Cd content in the northern sea area is the highest. The Cd content in the deep sea of the Arctic Ocean and the Canadian Basin are generally lower (Figure 3b).

The lowest content of Cr is located at the station BN07 in the southern entrance of the Bering Strait (the 5th CHINARE, 21.52 mg·kg^−1^); the highest content is 112.38 mg·kg^−1^, reached by the MS03 and M03 stations on both sides of the Chukchi Sea platform (the 4th of CHINARE). The Cr content near the Bering Strait is at a lower level compared with other sea areas, but the content of the Bering Sea is higher (Figure 3c).

The lowest content of Cu is located at BN05 station in the southern entrance of the Bering Strait (the 5th CHINARE, 3.40 mg·kg^−1^), and the highest content is located at BN10 station (the 4th CHINARE, 75.15 mg·kg^−1^) in the deep sea of the Arctic Ocean. The content of Cu in the Bering Sea, Chukchi Sea and the deep sea of the Arctic Ocean is higher (Figure 3d).

The lowest content of Ni is at NB04 station in the south Bering Sea (the 6th CHINARE, 17.1 mg·kg^−1^), the highest content is at SR12 station on the outer shelf of the Chukchi Sea (the 5th CHINARE, 144.91 mg·kg^−1^). The content of Ni in the deep sea of the Arctic Ocean, outer shelf of the Chukchi Sea, Beaufort Sea to Canadian Basin and south Bering Sea is relatively higher than other areas, but no obvious variation regularity is shown (Figure 3e).

The lowest content of Pb is located at CC03 station (the 5th CHINARE, 5.9 mg·kg^−1^) in the south Chukchi Sea shelf, the highest is at R11 station (the 6th CHINARE, 26.1 mg·kg^−1^). The Pb content in the Chukchi Sea, Beaufort Sea to Canadian Basin and the deep sea of the Arctic Ocean is obviously higher than that of the Bering Sea area, but no obvious variation regularity is shown (Figure 3f).

The lowest content of Zn is at SR05 station (the 4th CHINARE, 32.69 mg·kg^−1^) in the south Chukchi Sea shelf, and the highest content is at SR12 station (the 5th CHINARE, 166.86 mg·kg^−1^) on the outer shelf of Chukchi. The Zn content of the Bering Sea, Chukchi Sea, Canadian Basin and the deep sea of the Arctic Ocean is relatively higher than other areas, but no obvious variation regularity is shown (Figure 3g).

The lowest content of Hg is at BN05 station (the 5th CHINARE, 6.85 μg·kg^−1^) in the southern entrance of the Bering Strait; the highest content is at BL03 station (the 5th CHINARE, 140.98 × 10^−3^ mg·kg^−1^) in south Bering Sea. The Hg content in the southern entrance of the Bering Strait and the deep sea of Arctic Ocean is at a lower level, while the Hg content in the Beaufort Sea and outer shelf of Chukchi is higher and may have some degree of enrichment (Figure 3h).

Overall, as shown in Figure 4, in the study area, the ranges and differences of the trace metals are: As (13.85 ± 6.67 mg/kg), As (13.85 ± 6.67 mg/kg), Cd (0.20 ± 0.10 mg/kg), Cr (59.77 ± 17.74 mg/kg), Cu (23.08 ± 15.46 mg/kg), Ni (55.59 ± 26.72 mg/kg), Pb (13.52 ± 5.01 mg/kg), Zn (92.36 ± 27.64 mg/kg), THg[(38.07 ± 24.69) × 10^−3^ mg/kg].

The content of As, Cr, Cu, Ni and Zn in the sediments of the southern entrance of the Bering Strait are the lowest in the whole study area, except for Pb (Figure 4); the content of As, Cu, Ni, Pb, Zn and Hg in the Chukchi outer shelf are relatively high; in the deep sea of the Arctic Ocean, the enrichment of As, Cr, Cu, Ni, Pb and Zn are shown, especially As, while the content of Cd and Hg is obviously lower in this area; the content of Cr is much higher than other trace metals in the Beaufort Sea to the Canadian Basin area.

### 3.2. Geochemical Baseline Value of Trace Metals

The geochemical baseline values of As, Cd, Cr, Cu, Hg, Ni, Pb and Zn based on the relative cumulative frequency curve method are 14.31 mg kg^−1^, 0.23 mg kg^−1^, 55 mg kg^−1^, 27.5 mg kg^−1^, 0.037 mg kg^−1^, 68 mg kg^−1^, 14.2 mg kg^−1^ and 96 mg kg^−1^ (Figure 5). 

Table 7 shows the comparison of geochemical baseline values in this study and other literature.

Compared with element geochemical baseline values in the coastal Beaufort Sea and Kongsfjorden, the level of Cd, Ni, Zn and Hg are relatively different and the other four elements are closer.

### 3.3. Enrichment Factor (EF)

Table 8 shows the statistical result of the *EF* value of different elements at each sampling site.

According to the classification standard of *EF*, an *EF* value less than 2 means the enrichment of elements is mainly from natural input; an *EF* value greater than 2 means the enrichment of elements is related to human activity. The higher the *EF* value is, the greater the human interference is. According to the maximum value of *EF*, element As shows an obvious enrichment, with a maximum *EF* value of 9.88, followed by Cr element, with a maximum *EF* value of 3.08, indicating the moderate enrichment level of some sampling sites. The maximum *EF* values of the other six elements are all less than 2, indicating that the enrichment of these elements is not related to human activity (Figure 6).

Figure 7 and Figure 8 are the spatial distribution level of the *EF* values of As and Cr elements, respectively.

The maximum *EF* value of As is 9.88, located in the deep-sea area of the Arctic Ocean; the minimum value is 0.94, located in the south Bering Sea; the difference between the maximum and the minimum is 10.51. According to Figure 1, except for the south Bering Sea area, As shows anthropogenic enrichment in different degrees in six other regions, among which is the moderate enrichment in the South Chukchi Shelf, the outer shelf of Chukchi, the Beaufort Sea and the Canadian Basin; it also shows significant enrichment in the south Bering Sea and part of the deep-sea area of the Arctic Ocean.

The maximum *EF* value of Cr is 3.08, and the lowest value is 0.67, which is located on the outer shelf of Chukchi. The difference between the maximum value and the minimum value is 4.6. Only a few *EF* values are higher than 2, distributed in the Bering Sea shelf, the outer shelf of Chukchi and the Beaufort Sea-Canada Basin, indicating a lower anthropogenic enrichment level generally.

### 3.4. Geo-Accumulation Index (I_geo_)

Table 9 shows the statistical result of the *I_geo_* value of different elements at each sampling site.

According to Figure 9 and Figure 10, for the *I_geo_* value based on the geochemical baseline of each element, both the median and average value are less than 0, indicating an uncontaminated level. For the *I_geo_* based on the earth crust, the median and average value of As is 3, meaning a moderately contaminated to heavily contaminated level of As. The median and average value of Cr is between 1 and 2, meaning a moderately contaminated level of Cr.

The maximum *I_geo_* value of As is 4.68, located in the deep-sea area of the Arctic Ocean, meaning a heavily contaminated to extremely contaminated level of As in this area. The minimum value is 0.99, located in the South Bering Sea, and the *I_geo_* value is generally rising northward (Figure 11).

The maximum *I_geo_* value of Cr is 2.77, located on the outer shelf of Chukchi, meaning a moderately contaminated to heavily contaminated level. The minimum value is 0.38, located at the south entrance of the Bering Strait, and the *I_geo_* value in the north is higher than the south generally (Figure 12).

### 3.5. Potential Ecological Risk Index

On the basis of the formulas of the potential ecological risk index mentioned above, the $E_{R}^{i}$ and *RI* values of each element based on the baseline value and preindustrial reference value are shown in Table 10. In terms of average value, according to the $E_{R}^{i}$ based on the baseline value of the element, the potential ecological risk level of trace metals increases in the following sequence: Zn < Cr < Cu < Pb < Ni < As < Cd < Hg, and the sequence based on the preindustrial value is Zn < Pb < Cr < Cu < Ni < Cd < Hg < As. With the exception of Hg, which has an $E_{R}^{i}$ value of slightly over 40, all the other trace metals have $E_{R}^{i}$ values below 40, indicating a low potential risk level. The *RI* value range based on the baseline is 49.43–226.83, showing the potential risk level ranging from low to moderate, and the *RI* value range based on the preindustrial value is 14.29–54.70, showing a low potential risk level.

Figure 13 and Figure 14 are boxplots of two $E_{R}^{i}$ values. The Er median values based on the geochemical baseline of each element are all less than 40 (in which the Er median value of Hg element is close to 40), meaning the potential ecological risk of each single element is lower. The Er value based on the preindustrial value indicates an obviously much lower ecological risk of As element with all the median values less than 10.

As Figure 15 shows, the potential ecological risk indexes based on the geochemical baseline value (*RI*/baseline) in each site is much higher than those based on the preindustrial value (*RI*/preindustrial) proposed by Hakanson, while the changed trend is roughly similar. As far as the higher RI/baseline is concerned, only the RI/baseline values of five sites exceed 150 including B04, SR11, M07, BL03 and SR12, which are distributed in the South Bering Sea and the outer shelf of Chukchi and are at a moderate ecological risk level. According to the two *RI* values based on different reference values, the rest of the sites are all at a low ecological risk level.

## 4. Conclusions

In this study, the distribution characteristics and geochemical baselines of As, Cd, Cr, Cu, Ni, Pb, Zn and Hg in the surface sediment of the Pacific Sector of the Arctic Ocean and Bering Strait are explored; three methods are used to assess the ecological risks of them. The following points are concluded:(1)Cu, As and Hg had a little more content variation, and their contents in some areas were significantly higher than the previous reports.(2)Compared with previous studies, the content of Cr and Cu in some sites of the Bering Sea in this study are significantly higher than that of previous reports, the content of Pb showed a general higher trend, while the content of Cu and As of several sites in Chukchi continental shelf and the northern area are significantly higher than the previous reports.(3)According to the analysis of spatial distribution, the content of As, Cr, Cu, Ni and Zn in the sediments of the southern entrance of the Bering Strait are the lowest in the whole study area, except for Pb; the content of As, Cu, Ni, Pb, Zn and Hg in the Chukchi outer shelf are relatively higher; in the deep sea of the Arctic Ocean, the content of As, Cr, Cu, Ni, Pb and Zn are higher, while Cd and Hg are obviously lower; the content of Cr is much higher than others in the Beaufort Sea to Canadian Basin area.(4)According to the maximum value of *EF*, element As shows an obvious enrichment with a maximum *EF* value of 9.88, followed by Cr element, with a maximum *EF* value of 3.08, indicating the moderate enrichment level of some sampling sites. The maximum *EF* values of the other six elements are all less than 2, indicating that the enrichment of these elements is not related to human activity.(5)For the *I_geo_* value based on the geochemical baseline of each element, both the median and average value are less than 0, indicating an uncontaminated level. For the *I_geo_* based on the earth crust, As and Cr show a relatively obvious contaminated level.(6)The potential ecological risk indexes based on the geochemical baseline value in each site are much higher than those based on the preindustrial value proposed by Hakanson, while the changed trend is roughly similar. Only the RI/baseline values of five sites exceed 150, meaning a moderate ecological risk level, the rest of the sites are all at a low ecological risk level.

## Figures and Tables

**Figure 1 ijerph-19-04454-f001:**
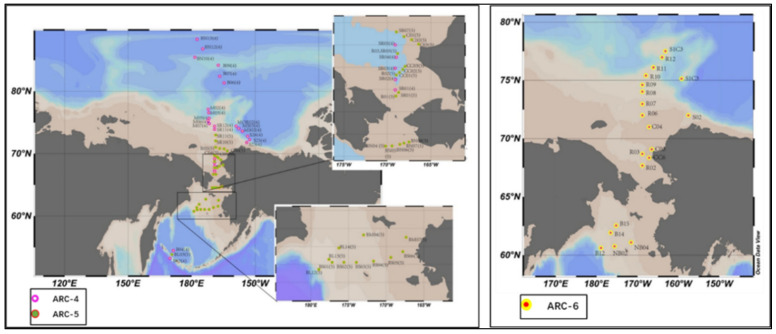
Location of study area and sampling sites.

**Figure 2 ijerph-19-04454-f002:**
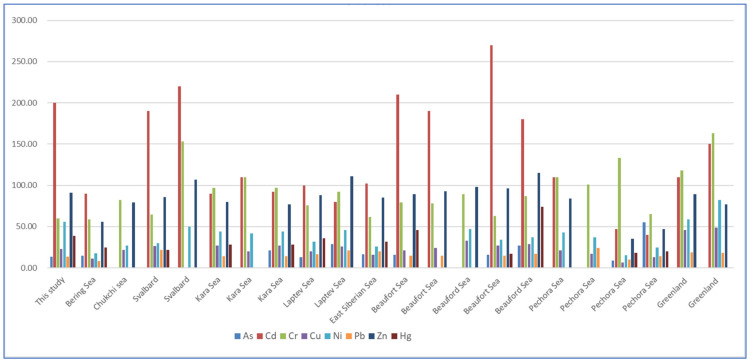
Comparison of trace metal content with other Arctic areas.

**Figure 3 ijerph-19-04454-f003:**
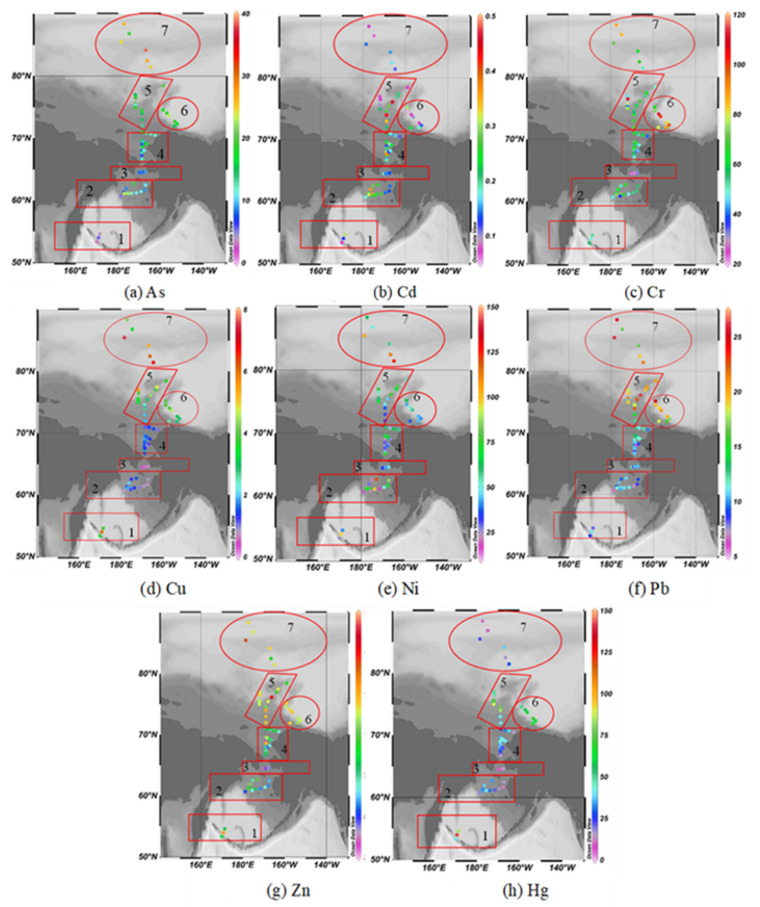
Spatial distribution of trace metal (**a**) As, (**b**) Cd, (**c**) Cr, (**d**) Cu, (**e**) Ni, (**f**) Pb, (**g**) Zn and (**h**) Hg in study area. Note: 1. South Bering Sea; 2. the Bering Sea shelf; 3. the southern entrance of Bering Strait; 4. the south Chukchi Sea shelf; 5. the outer shelf of Chukchi; 6. Beaufort Sea to the Canadian Basin; 7. the deep-sea area of the Arctic Ocean.

**Figure 4 ijerph-19-04454-f004:**
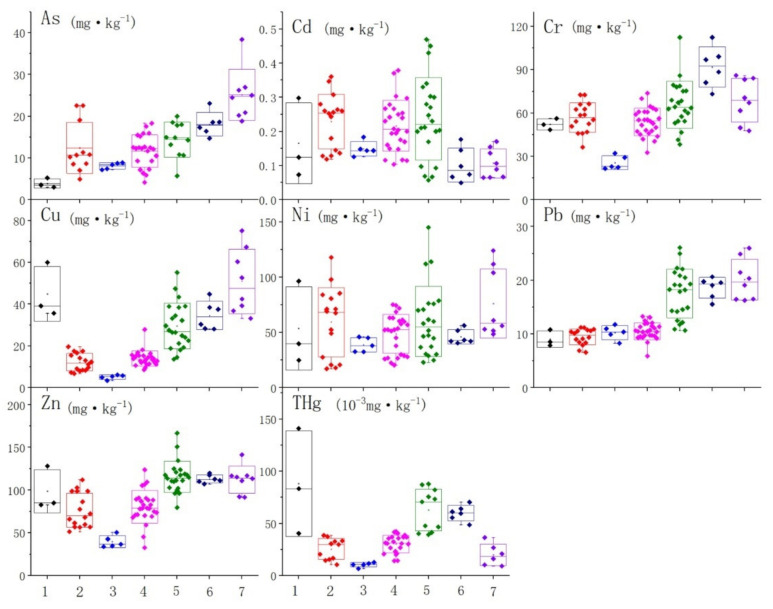
Content distribution of trace metals in each portion. Note: 1. South Bering Sea; 2. the Bering Sea shelf; 3. the southern entrance of Bering Strait; 4. the south Chukchi Sea shelf; 5. the outer shelf of Chukchi; 6. Beaufort Sea to the Canadian Basin; 7. the deep-sea area of the Arctic Ocean.

**Figure 5 ijerph-19-04454-f005:**
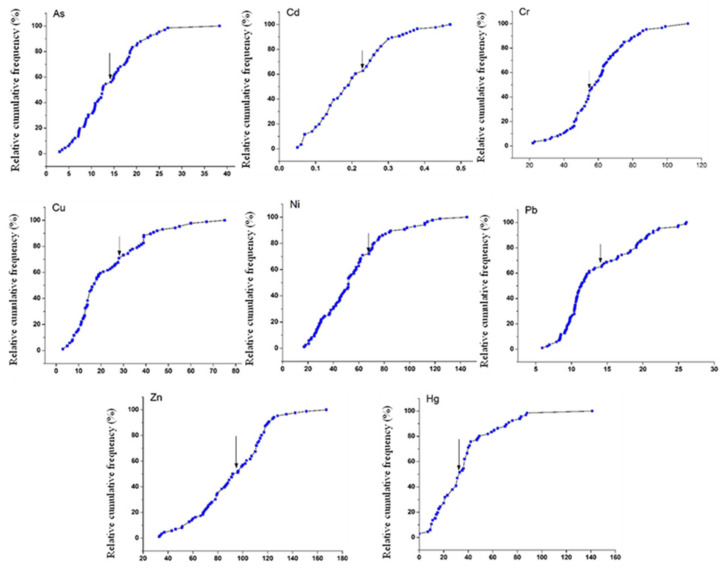
Geochemical baseline values of trace metals based on the relative cumulative frequency curve.

**Figure 6 ijerph-19-04454-f006:**
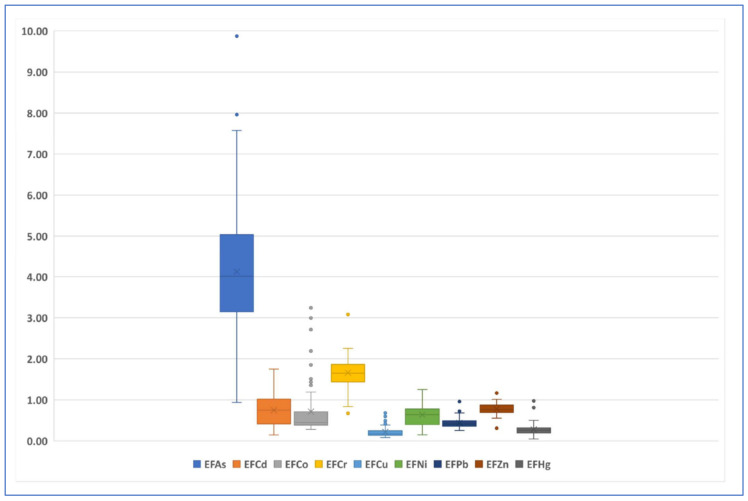
Boxplot of *EF* value of each element.

**Figure 7 ijerph-19-04454-f007:**
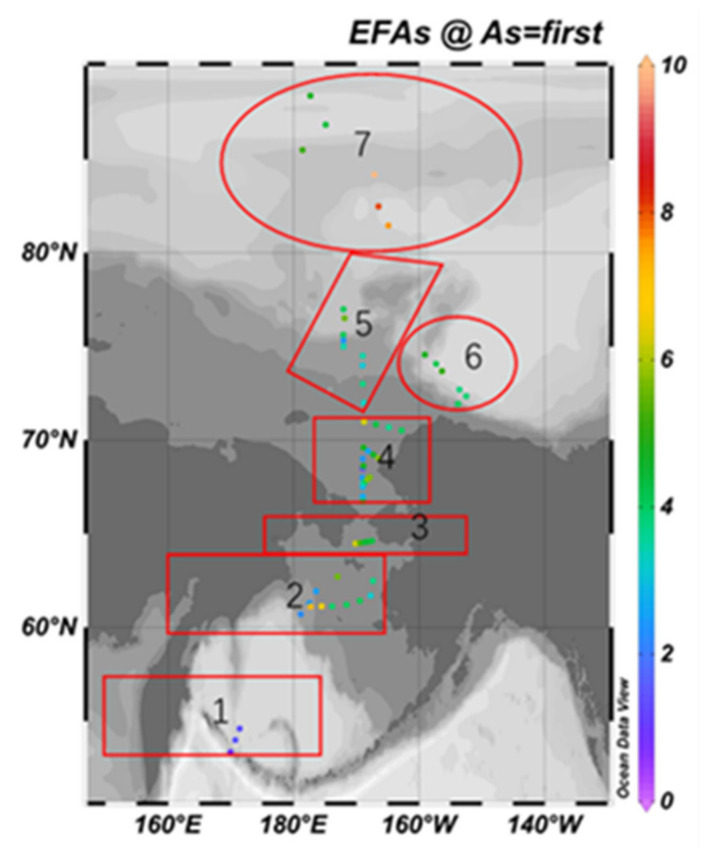
Spatial distribution of *EF* value of As element.

**Figure 8 ijerph-19-04454-f008:**
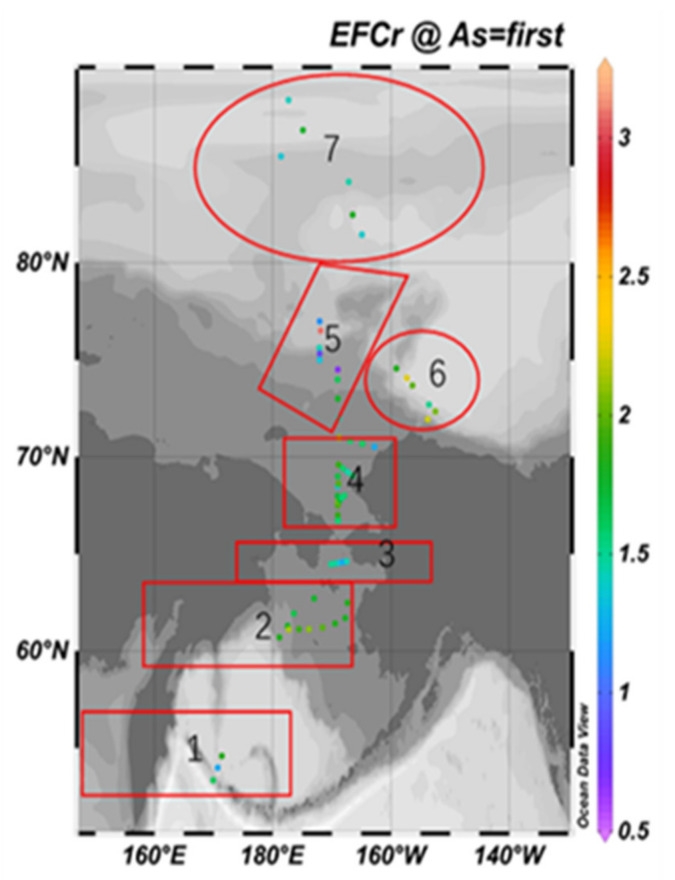
Spatial distribution of *EF* value of Cr element.

**Figure 9 ijerph-19-04454-f009:**
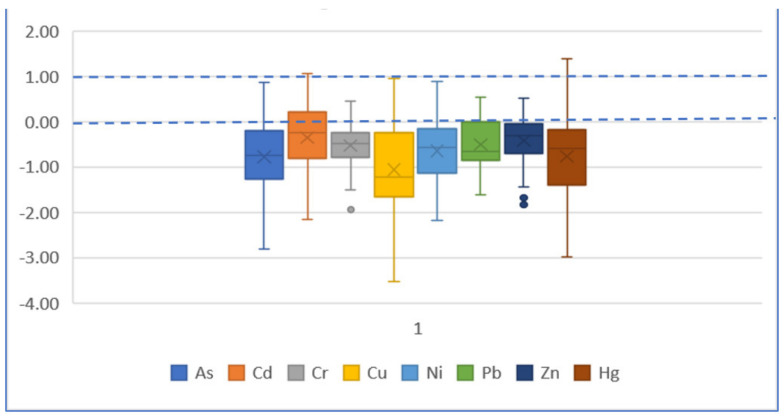
Boxplot of *I_geo_* based on baseline value for each element.

**Figure 10 ijerph-19-04454-f010:**
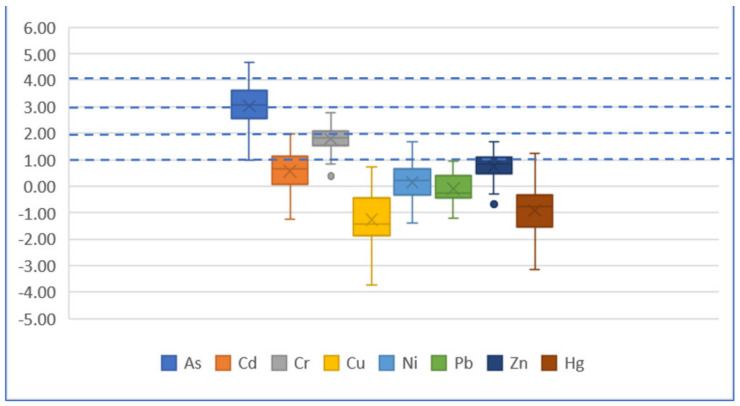
Boxplot of *I_geo_* based on earth crust for each element.

**Figure 11 ijerph-19-04454-f011:**
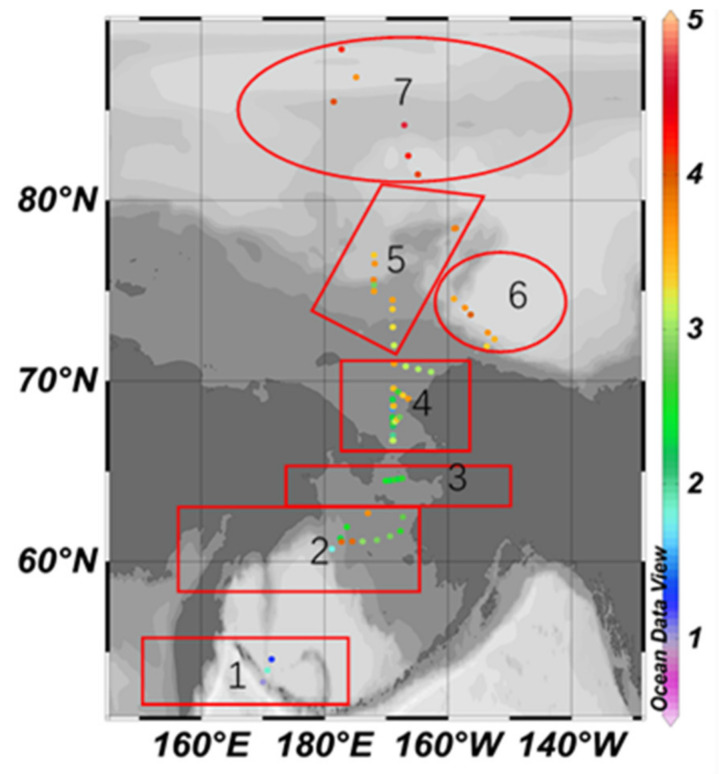
Spatial distribution of *I_geo_* based on earth crust for As element.

**Figure 12 ijerph-19-04454-f012:**
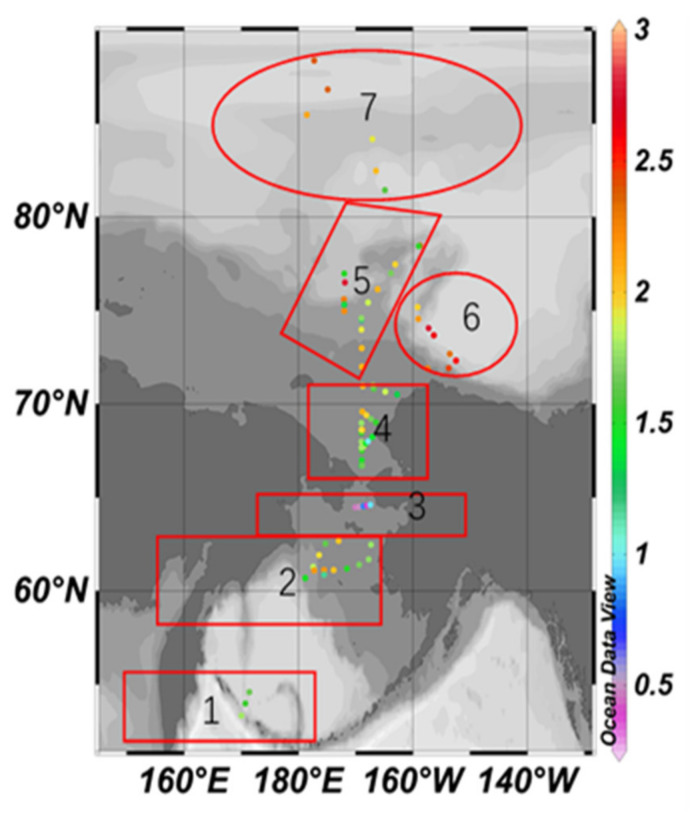
Spatial distribution of *I_geo_* based on earth crust for Cr element.

**Figure 13 ijerph-19-04454-f013:**
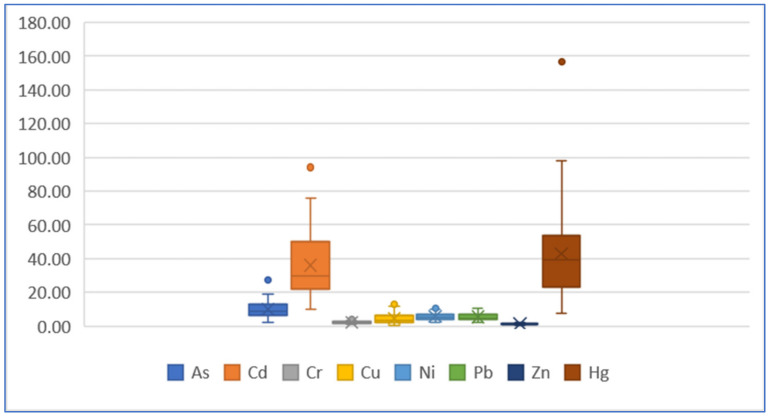
Boxplot of $E_{R}^{i}$ value of each element based on baseline.

**Figure 14 ijerph-19-04454-f014:**
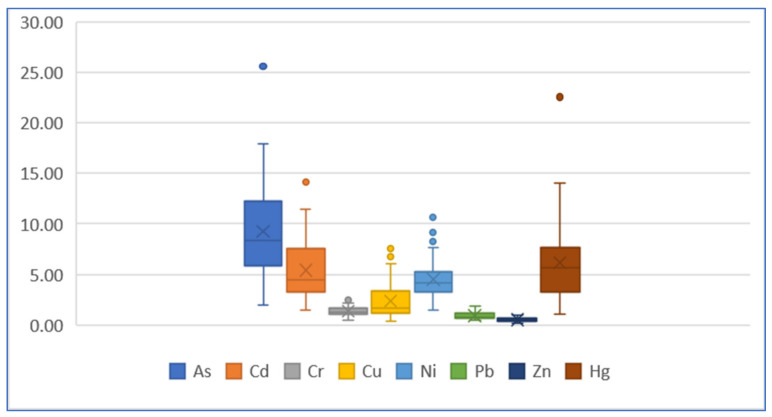
Boxplot of $E_{R}^{i}$ value of each element based on preindustrial value.

**Figure 15 ijerph-19-04454-f015:**
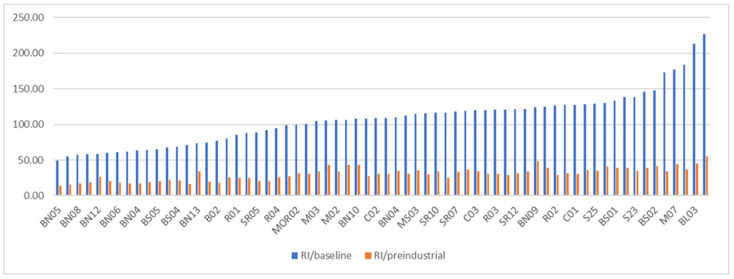
*RI* value based on baseline and preindustrial value.

**Table 1 ijerph-19-04454-t001:** Concentration of trace metals in the earth crust [21]. Unit: mg/kg.

Element	As	Cd	Cr	Cu	Ni	Pb	Zn	Hg	Fe
Earth crust	1	0.08	11	30	30	9	35	0.04	9000

**Table 2 ijerph-19-04454-t002:** Classification standard of *EF* value.

*EF* Value	Class	Classification
<2	1	Deficiency to minimal enrichment
2–5	2	Moderate enrichment
5–20	3	Significant enrichment
20–40	4	Very high enrichment
>40	5	Extremely high enrichment

**Table 3 ijerph-19-04454-t003:** Classification of Igeo value.

Class	*I_geo_* Value	Classification
0	<0	uncontaminated
1	0~1	uncontaminated to moderately contaminated
2	1~2	moderately contaminated
3	2~3	moderately contaminated to heavily contaminated
4	3~4	heavily contaminated
5	4~5	heavily contaminated to extremely contaminated
6	5~6	extremely contaminated

**Table 4 ijerph-19-04454-t004:** Preindustrial reference value of trace metals. Unit: mg/kg.

Element	As	Cd	Cr	Cu	Ni	Pb	Zn	Hg
Preindustrial reference	15	1	90	50	68	70	175	0.25

**Table 5 ijerph-19-04454-t005:** Classification of $E_{R}^{i}$ and *RI*.

$E_{R}^{i}$ Value	Description	*RI* Value	Description
$E_{R}^{i}$ < 40	low potential ecological risk	*RI* < 150	low potential ecological risk
40 < $E_{R}^{i}$ < 80	moderate potential ecological risk	150 < *RI* < 300	moderate ecological risk
80 < $E_{R}^{i}$ < 160	considerable potential ecological risk	300 < *RI* < 600	considerable ecological risk
160 < $E_{R}^{i}$ < 320	high potential ecological risk	*RI* > 600	very high ecological risk
$E_{R}^{i}$ < 320	very high ecological risk		

**Table 6 ijerph-19-04454-t006:** The statistical distribution of trace metals in surface sediment of study area. Unit: mg/kg.

Trace Metal	As	Cd	Cr	Cu	Ni	Pb	Zn	Hg
Max	38.34	0.47	112.38	75.15	144.91	26.1	166.86	0.14
Min	2.98	0.05	21.52	3.4	17.1	5.9	32.69	0.007
Average	13.85	0.20	59.77	23.08	55.59	13.52	90.70	0.039
Median	12.50	0.19	58.58	16.75	52.26	11.38	94.24	0.03
Max/min	12.87	9.40	5.22	22.10	8.47	4.42	5.10	20.58
Skewness	0.89	0.59	0.52	1.24	1.00	0.86	−0.15	1.51

**Table 7 ijerph-19-04454-t007:** Comparison of geochemical baseline value with other Arctic area results. Unit: mg/kg.

Metal	As	Cd	Cr	Cu	Ni	Pb	Zn	Hg
This study	14.31	0.23	59.75	27.5	68	14.2	96	0.037
Coastal Beaufort Sea [31]	12.2		82.5	30.2	39.39	14.1	117.5	0.066
Kongsfjorden, Svalbard, Norwegian Arctic [32]		0.14	57.86	25.14	26.22	17.46	70.49	0.0976

**Table 8 ijerph-19-04454-t008:** The statistical result of *EF* value of each element.

*EF*	As	Cd	Cr	Cu	Ni	Pb	Zn	Hg
MAX	9.88	1.75	3.08	0.68	1.25	0.96	1.17	0.97
MIN	0.94	0.15	0.67	0.08	0.15	0.25	0.31	0.05
AVE	4.17	0.75	1.66	0.22	0.64	0.44	0.78	0.28

**Table 9 ijerph-19-04454-t009:** Statistical analysis of *I_geo_* based on baseline value and earth crust for each element.

Trace Metal		As	Cd	Cr	Cu	Ni	Pb	Zn	Hg
*I_geo_*/baseline	MAX	0.87	1.06	0.45	0.95	0.89	0.54	0.51	1.38
	MIN	−2.82	−2.17	−1.94	−3.52	−2.19	−1.61	−1.84	−2.98
	AVE	−0.78	−0.35	−0.53	−1.06	−0.65	−0.50	−0.42	−0.76
*I_geo_*/crust	MAX	4.68	1.97	2.77	0.74	1.69	0.95	1.67	1.23
	MIN	0.99	−1.26	0.38	−3.73	−1.40	−1.19	−0.68	−3.13
	AVE	3.03	0.56	1.79	−1.27	0.14	−0.09	0.74	−0.92

**Table 10 ijerph-19-04454-t010:** Statistical result of $E_{R}^{i}$ and *RI* value.

			Min	Ave	Max
$E_{R}^{i}$	As	$E_{R}^{i}$/baseline	2.13	9.89	27.39
		$E_{R}^{i}$/preindustrial	1.99	9.23	25.56
	Cd	$E_{R}^{i}$/baseline	10.00	35.86	94.00
		$E_{R}^{i}$/preindustrial	1.50	5.38	14.10
	Cr	$E_{R}^{i}$/baseline	0.78	2.20	4.09
		$E_{R}^{i}$/preindustrial	0.48	1.34	2.50
	Cu	$E_{R}^{i}$/baseline	0.65	4.51	14.45
		$E_{R}^{i}$/preindustrial	0.34	2.35	7.52
	Ni	$E_{R}^{i}$/baseline	1.97	5.87	13.93
		$E_{R}^{i}$/preindustrial	1.51	4.49	10.66
	Pb	$E_{R}^{i}$/baseline	2.46	5.46	10.82
		$E_{R}^{i}$/preindustrial	0.42	0.94	1.86
	Zn	$E_{R}^{i}$/baseline	0.42	1.16	2.14
		$E_{R}^{i}$/preindustrial	0.19	0.52	0.95
	Hg	$E_{R}^{i}$/baseline	7.61	42.86	156.64
		$E_{R}^{i}$/preindustrial	1.10	6.17	22.56
*RI*		*RI*/baseline	49.43	107.81	226.83
		*RI*/preindustrial	14.29	30.42	54.70

## Data Availability

Not applicable.

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
