# Peer review of "Ecological Risk Assessment of Trace Metal in Pacific Sector of Arctic Ocean and Bering Strait Surface Sediments"

_ijerph, 2022, doi:10.3390/ijerph19084454_

Round 1

Reviewer 1 Report

The paper presents the results of the content of heavy metals in sediments collected in the Pacific Sector of the Arctic Ocean and Bering Strait. There already are a number a studies concerning this topic; which is the novelty of this study? Which kinds of information does it bring over the existing state of the art?

First of all, it is not clear from Figure 1 if sediments are collected on the ground (in soil) or at the bottom of the ocean in zones of shallow waters or in both locations.

Methods used for the determination of the geochemical baseline (section 2.4) and the ecological risk assessment (section 2.5) are not adequately described. Not clear the source of the data reported in Table 1.

Lines 90-93 – the sentence here reported is not clear  

There is not agreement between the information reported in line 80 (the sediments were stored in glass bottles) and 95 (the sediments were stored in LDPE bottles).

Line 99 – details concerning the preparation of samples and grinding must be reported.

Line 103 – not clear to me the reason why the sample was firstly dried to be pretreated but the weight of the sample subjected to digestion was wet. Details concerning the digestion (i.e. the temperature of the process or the power of microwave) must be reported.

Line 114 – not clear how many replicas of digestion or digestate ICP-MS analyses were performed.

Is there a relation between Table 7 and Figure 2?

In general, the quality of the Figures must be improved, because most of them (i.e. 3, 7, 8, 11 and 12) are not clearly legible.

Which kind of data does it report Figure 4? Only those from this study or even other data coming from other studies?

Figure 5 – not completely clear the way in which diagrams were build (were all the data collected in this study used?)

The sections concerning the geo-accumulation index and the potential ecological risk index must be expanded. Which is the implication of these results?

Finally, English is not my first language but I feel that the quality of the language and sentence structure must be improved with the help of a professional native English speaker proof-reader.

Author Response

Dear Reviewer:

Please find attached revise version of your kind comments.

Reviewer 2 Report

General comment

The study “Ecological risk assessment of trace metal in Pacific Sector of Arctic Ocean and Bering Strait Sediments” explain the spatial distribution of 8 trace metals in Arctic Ocean and their potential ecological risks. Although the study made an effect to reveal the difference between sampling sites, the weakness is mainly related to statistical methods. Therefore, I recommend the present article to be published in the journal of “International Journal of Environmental Research and Public Health” after major revision.

Specific Comments

  1. Please more about the importance of polar research.
  2. There are so many figures on manuscript. It’s better to refine.
  3. P30: Please, give a full name of seven elements (As, Cd, Cr, Cu, Ni, Pb, and Zn).
  4. P95–97: Please, give a full name of LDPE and seven elements (As, Cd, Cr, Cu, Ni, Pb, and Zn).
  5. Figure 4, Please, give statistical results for these differences of distributions.

Author Response

Dear Reviewer:

Please find attached version of my paper.

Round 2

Reviewer 1 Report

The quality of the presentation has improved after review. The authors replied to all my comments.

Reviewer 2 Report

The author has modified the manuscript according to reviewer's comments. I agree it can be accepted.